# Displacement Monitoring Requirements and Laser Displacement Monitoring Technology of Bridges with Short and Medium Spans

**Xiaoya Li** [1,2] **and Fangyuan Li** [3,*]

1 Key Laboratory of Road and Bridge Detection and Maintenance Technology of Zhejiang Province, Hangzhou 311305, China
2 Zhejiang Scientific Research Institute of Transport, Hangzhou 310023, China
3 Department of Bridge Engineering, College of Civil Engineering, Tongji University, Shanghai 200092, China
* Correspondence: fyli@tongji.edu.cn

**Abstract:** Short and medium-span bridges account for more than 90% of the bridges in China, and it is not suitable to use conventional health monitoring technology to monitor them. Based on an analysis of the proportion of live load, displacement under load, the action time of live load and other factors affecting short and medium-span bridges, this paper determines a feasible technical solution using mid-span deflection to monitor bridges with requirements including measuring range, resolution and accuracy. A set of wireless laser displacement monitoring technologies and equipment is studied by using the phase laser ranging method and the principle of pulsed laser ranging, and the reliability of the data is verified by experimental tests. Using both practical application cases and economic analysis, this paper proves that the technology has significant application advantages for short and medium-span bridges.

**Keywords:** girder bridge; mid-span deflection; load-displacement relationship; laser ranging; health monitoring system

## 1. Introduction

In normal operation and traffic processes, various external loads such as traffic, pedestrians, and temperature act on a bridge, and these loads will have different effects on the safety and reliability of the bridge structure. To track the performance and assess the safety of a bridge, researchers typically use field tests and numerical simulation to obtain its health status [1,2]. Bridge deflection refers to the vertical displacement changes of bridges under the action of loads, and these changes have a great impact on the safety characteristics of bridges [3,4]. Therefore, the monitoring of bridge displacement is an important part of the health monitoring system, which is of great significance to the safe operation of bridges. It is known that deflection is one key factor used to evaluate structural performance since bridge deflection under the same level load is mainly caused by stiffness change, in other words, under the same level of load, if structure stiffness is reduced and the deformation will increase. Miyamoto used characteristic deflection as a factor in developing a long-term health monitoring system for short and medium-span bridges, which is relatively free from the influence of dynamic disturbances due to such factors as the roughness of the road surface, and a structural anomaly parameter [5]. Although there is not a linear relation between damage and deflection for many factors' impacts, at least there is a relationship in some degree. In view of long-term use, overload, impact and other factors, the cracks in concrete beam structures continue increasing in China's bridges with heavy traffic flow; an increase in deflections has also been shown in these bridges.

There is a huge amount of short and medium-span bridges in the world [6]. For example, by the end of 2019, the total number of highway bridges in Zhejiang Province,

China, had reached 51,106, including 46,000 short and medium-span bridges, accounting for 90% of the total [7]. Among them, hollow slab girder bridges, T-beam bridges and arch bridges accounted for the vast majority of short and medium-span bridges. Short and medium-span bridges have relatively short spans and small deflections compared with long-span bridges such as cross-sea and river-crossing bridges because the monitoring of dynamic deflection for long-span bridges has higher requirements on the hardware and software of the system [8]. At present, the prices of the existing products on the market are relatively expensive. On the other hand, the installation always takes a long time [9–11]. As a result, these health monitoring systems for short and medium-span bridges have not been widely promoted and applied [12].

Due to the great differences in material, span, environment, etc., a variety of detection principles, measurement methods and instruments for bridge deflection detection have been researched and developed at home and abroad. Bridge deflection measurement methods mainly include the dial indicator method, level gauge measurement method, theodolite measurement method, total station measurement method, inclinometer measurement method, digital imaging or photoelectric imaging measurement method, connecting pipe measurement method, radar interferometry and GPS measurement method, etc. [8,10,13–15].

The application range and measurement accuracy of the above displacement sensors are shown in Table 1.

**Table 1.** Comparison of different displacement sensors.

| ID | Method | Accuracy | Scope of Application | Number of Points | Benchmark | Results Representation | Application |
|----|--------|----------|----------------------|------------------|-----------|------------------------|-------------|
| 1 | dial indicator | um | SD | single or multiple points/each time | GD | changes relative to the earth | commonly |
| 2 | level | mm | SD | single point/each time | GD | changes from horizontal line of sight | commonly |
| 3 | theodolite | cm | SD | single point/each time | GD | changes from horizontal line of sight | commonly |
| 4 | total station | mm | SD | single point/each time | IP | changes from the initial line of sight | commonly |
| 5 | inclinometer | cm | SD/DD | single or multiple points/each time | IR | change from initial state | not much |
| 6 | photo-electric imaging | mm~cm | SD/DD | single point/each time | OA | change from initial position | commonly |
| 7 | connecting pipe | mm | SD | single or multiple points/each time | HP | change from initial state | commonly |
| 8 | differential GPS method | cm | SD/DD | single or multiple points/each time | GD | variation with respect to geodetic coordinates | commonly |
| 9 | laser method | mm | SD/DD | single point/each time | CL | variation relative to collimated laser beam | not much |

Note: scope of application: SD—static deflection, DD-low frequency dynamic; benchmark: GD—geodetic datum, IP—initial position line of sight reference, IR—inertial reference, OA—optical axis reference of imaging system, HP—horizontal plane. CL—collimated laser beam reference.

The commonly used deflection monitoring equipment for bridge health monitoring mainly includes connecting pipes, satellite positioning systems and inclinometers. Judging from the existing bridge deflection measurement equipment, there are mainly the following deficiencies in the application of short and medium-span bridge structure monitoring [3,9,13,16]. First, monitoring is a long-term, real-time, and uninterrupted process, and some measurement equipment cannot work under all weather and in real time. Second, there are a large number of hollow slabs and some T-shape beam structures in short-span or medium-span bridges, and measurement equipment is not easy to install and arrange, especially for bridges across rivers and lines. Third, the general span of short and medium-span bridges and the deformation is small, and the measurement accuracy and sampling frequency of deflection are relatively small. Fourth, the cost of some sensors is relatively high, which is not suitable for short and medium-span bridge

monitoring. Therefore, it is necessary to upgrade the existing deflection monitoring products to reduce costs and simplify on-site installation.

## 2. Analysis of the Requirements for Displacement Monitoring Indicators of Short and Medium Span Bridges

In this paper, some short and medium-span, simply supported, girder bridges in practical engineering were selected to establish a finite element calculation model to study the constant live load ratio [11,16]. Using Midas Civil software, the finite element model of the traditional beam element was analyzed according to the Chinese highway bridge design code and the general load code. The software automatically loaded the influence line according to the layout lane based on the Pingli Highway project in Jiashan Town, Zhejiang Province, which is a first-class highway with two lanes. The design load grade is autoload-20 and trailer-load-100. The upper structure of the bridge is mainly composed of reinforced concrete and post-tensioned, prestressed, concrete hollow slabs [17]. The key parameters of the simply supported, hollow slab bridge on Pingli Highway are shown in Table 2.

**Table 2.** Key parameters of the simply supported hollow slab bridge on the Pingli Highway.

| Structure Type | Sectional Form | Span (m) | Beam Height (m) | Board Width (m) | Number of Single Bridge Pieces |
|---|---|---|---|---|---|
| rebar concrete |  | 8 | 0.4 | 1.25 | 9 |
| | | 10 | 0.45 | 1.25 | 10 |
| prestressed concrete |  | 13 | 0.55 | 1 | |
| | | 16 | 0.80 | 1 | 9 |
| | | 20 | 0.90 | 1 | 11 |
| | | 25 | 1.10 | 1 | 13 |
| | | 30 | 1.20 | 1 | |
| | | 29.60 30.20 | 1.40 | 1 | 13 |
| | | 33.95 | 1.40 | 1 | |

We extracted the constant and live load bending moment values at the mid-span positions of the abovementioned sample bridges (prestressed hollow slab girder bridges do not temporarily consider the effect of prestressing) and considered the normal use in structural design and the load combination in bearing capacity calculation [18,19], respectively. The proportion of the live load effect to the total effect of the constant live load under the standard action combination and the bearing capacity limit combination is recorded as $\eta_1$ and $\eta_2$. The specific calculation is shown in the following Formulas (1) and (2):

$$\eta_1 = \frac{\varphi_{11} M_{Q1}}{\varphi_{11} M_{Q1} + \varphi_{12} M_G} \times 100\% \tag{1}$$

$$\eta_2 = \frac{\varphi_{21} M_{Q1}(1+\mu)}{\varphi_{21} M_{Q1}(1+\mu) + \varphi_{22} M_G} \times 100\% \tag{2}$$

In the formula, $\varphi_{11}$ and $\varphi_{12}$ are the corresponding coefficients of the live and dead load effects of the standard combination, ($\varphi_{11} = \varphi_{12} = 1.0$ respectively) $\varphi_{21}$ and $\varphi_{22}$ are the corresponding coefficients of the live and dead load effects of the ultimate combination of bearing capacity ($\varphi_{21} = 1.4$, $\varphi_{22} = 1.2$ respectively). $M_{Q1}$ is the bending moment value

generated by the living load; $M_G$ is the bending moment value generated by the dead load; and $\mu$ is the load impact system of the vehicle [18].

The results of calculating the live load proportion of the sample bridges are shown in Table 3.

**Table 3.** The constant live load ratio of common short and medium bridges on Pingli Highway.

| Structural Properties | | | Maximum Mid-Span Bending Moment of Main Beam (kN·m) | | | | $\eta_1$ (%) | $\eta_2$ (%) |
|---|---|---|---|---|---|---|---|---|
| | | | Dead Load | Live Load | Standard Combination | Bearing Capacity Limit State Combination | | |
| Structure Type | Span (m) | Beam Plate Quantity (pieces) | Middle Beam | Middle Beam | Middle Beam | Middle Beam | | |
| ordinary reinforced concrete bridge | 8 | 9 | 72.3 | 240.3 | 312.6 | 423.2 | 76.9 | 79.5 |
| | | 10 | 72.3 | 240.2 | 312.5 | 423.0 | 76.9 | 79.5 |
| prestressed concrete bridge | 13 | 9 | 181.2 | 263.0 | 443.1 | 584.4 | 59.4 | 62.8 |
| | | 11 | 181.2 | 256.0 | 436.2 | 574.6 | 58.7 | 62.2 |
| | | 13 | 181.3 | 253.2 | 433.3 | 570.6 | 58.4 | 61.9 |
| | 16 | 9 | 344.3 | 353.0 | 695.8 | 905.3 | 50.7 | 54.4 |
| | | 11 | 344.4 | 339.3 | 683.3 | 887.7 | 49.7 | 53.5 |
| | | 13 | 344.4 | 333.9 | 678.3 | 880.8 | 49.2 | 53.1 |
| | 20 | 9 | 617.2 | 456.3 | 1066.8 | 1371.4 | 42.8 | 46.0 |
| | | 11 | 617.5 | 450.5 | 1066.8 | 1370.0 | 42.2 | 45.9 |
| | | 13 | 617.7 | 437.7 | 1054.6 | 1352.9 | 41.5 | 45.2 |
| | 25 | 9 | 1117.5 | 646.4 | 1762.1 | 2243.8 | 36.7 | 40.2 |
| | | 11 | 1118.1 | 597.5 | 1713.6 | 2175.8 | 34.9 | 38.3 |
| | | 13 | 1118.5 | 573.9 | 1690.4 | 2143.2 | 34.0 | 37.4 |
| | 30 | 9 | 1728.8 | 611.8 | 2340.6 | 2931.1 | 26.1 | 29.2 |
| | | 11 | 1730.0 | 738.0 | 2467.5 | 3108.5 | 29.9 | 33.2 |
| | | 13 pieces | 1730.8 | 686.1 | 2416.9 | 3037.5 | 28.4 | 31.6 |

From Table 3 it can be seen that the proportion of the live load effect of common short and medium-span bridges on highways is large. The shorter the span is, the larger the proportion of live load. As the bridge width increases, the proportion of live load also decreases slightly. Therefore, in the structural monitoring of short and medium-span bridges, the live load effect will occupy a very important position [20,21]. It is necessary to adopt a dynamic monitoring system with the right sampling frequency and supporting sensors to ensure the integrity of the actual monitoring data and to capture various deflection effects caused by live loads, which is conducive to the evaluation of the operating state of the bridge. Therefore, for the deflection of short and medium-span bridges, it is of great significance to use dynamic monitoring for displacement monitoring.

For short and medium-span bridges, if the vehicle speed is 60 km/h, the time required for vehicles to pass over bridges with different spans is shown in Table 4.

**Table 4.** The time it takes for vehicles to pass over bridges of different spans (unit: seconds).

| Span | Vehicle Passing Time | Span | Vehicle Passing Time |
|---|---|---|---|
| 8 m | 0.48 s | 20 m | 1.2 s |
| 10 m | 0.6 s | 25 m | 1.5 s |
| 13 m | 0.78 s | 30 m | 1.8 s |
| 16 m | 0.96 s | / | / |

It can be observed from Table 4 that when vehicles pass over bridges with short and medium spans, the time is short. To completely record the maximum deflection of vehicles when vehicles pass over bridges and comprehensively consider economy, the sampling frequency of dynamic deflection and displacement monitoring of short and medium-span bridges is recommended to be no less than 10 Hz.

The deformation of common short and medium-span bridges on Pingli Highway under live load is shown in Table 5.

**Table 5.** Deformation of bridges under live loads on Pingli Highway.

| Structure Type | Span (m) | Beam Plate Quantity (pieces) | Mid-Span Deformation of Side Beam (mm) |
|---|---|---|---|
| ordinary reinforced concrete bridge | 8 | 9 | 5.74 |
| | | 10 | 5.69 |
| prestressed concrete bridge | 13 | 9 | 7.25 |
| | | 11 | 6.85 |
| | | 13 | 6.29 |
| | 16 | 9 | 5.80 |
| | | 11 | 5.64 |
| | | 13 | 4.94 |
| | 20 | 9 | 9.30 |
| | | 11 | 9.21 |
| | | 13 | 8.17 |
| | 25 | 9 | 11.90 |
| | | 11 | 11.62 |
| | | 13 | 10.22 |
| | 30 | 9 | 12.60 |
| | | 11 | 16.36 |
| | | 13 | 14.27 |

In Table 5, the deformation of the main beam under the action of the vehicle on the mid-span is 5.74 mm (minimum) and 14.27 mm (maximum). Considering the margin of deformation, the range of the laser displacement monitoring equipment is designed to be ±30 mm, the resolution is 0.1 mm, and the accuracy is ±0.3 mm.

In summary, the basic performance parameters of the laser displacement monitoring equipment are shown in Table 6.

**Table 6.** Basic performance requirements parameters of laser displacement monitoring equipment.

| Precision | Resolution (mm) | Range (mm) | Protection Class | Sampling Frequency (Hz) |
|---|---|---|---|---|
| ±0.3 mm | 0.1 | ±30 | IP67 | 10 Hz |

Note: IP—Ingress Protection Rating, IP67—waterproof standard should be 67 Grade.

### 3. Development of Wireless Laser Displacement Monitoring Equipment

*3.1. The Principle of Laser Ranging Measurement*

At present, there are two main types of laser ranging methods: time-of-flight (TOF) ranging and non-time-of-flight (NTOF) ranging. Two methods are mainly used for TOF, including phase laser ranging and pulse laser ranging, while triangular laser ranging is mainly used for NTOF. This paper mainly uses the principle of TOF ranging to develop wireless laser displacement monitoring equipment [20,22–25].

#### 3.1.1. Phase Laser Ranging

Figure 1 shows that the phase ranging method continuously modulates the amplitude of the emitted laser, usually using sine wave modulation. Since the delay between the laser round-trip ranging system and the target will produce a phase change, the phase of the delay between the receiving and transmitting optical signals is measured. Since the frequency of the modulating signal is known, the laser beam can calculate flight time and finally calculate the distance to the target according to the formula [26].

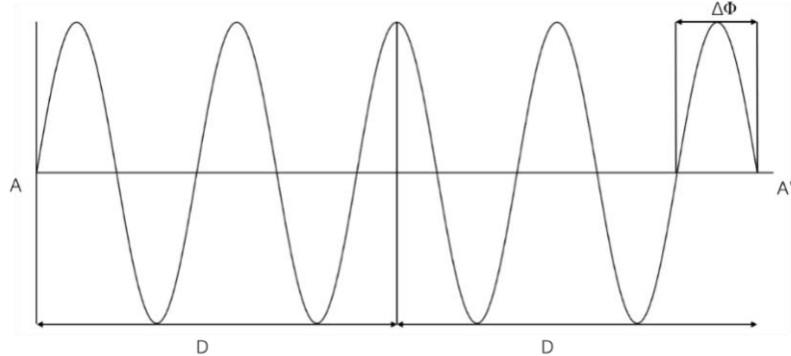

**Figure 1.** Schematic diagram of phase ranging.

The distance $D$ satisfies Formula (3):

$$D = \frac{ct}{2} \tag{3}$$

$$t = \frac{\varphi}{\omega} = \frac{2\pi N + \Delta\varphi}{2\pi f}(N + \frac{\Delta\varphi}{2\pi})\frac{1}{f} \tag{4}$$

Put Formula (4) into Formula (3) to obtain Formula (5):

$$D = \frac{\lambda c}{2}(N + \frac{\Delta\varphi}{2\pi}) \tag{5}$$

where $c$ is the speed of light in the air; $f$ and $\lambda$ are the frequency and wavelength of the modulated signal, respectively; $\Delta\varphi$ and $N$ are the phase differences caused by the marking-modulated optical signal flying back and forth between the ranging system and the target to be measured, respectively; $N$ is the integer part of one phase difference; and $\Delta\varphi$ is the mantissa of the unsatisfied integer period.

The half wavelength of the modulation signal is $\frac{\lambda}{2}$ defined as the measuring ruler; then, the phase laser ranging can be expressed as using a measuring range as the $\frac{\lambda}{2}$ ruler to measure the distance. Formulas (4) and (5) can be obtained by a certain method, but $N$ is difficult to obtain by a simple method, so the distance $D$ obtained by Formula (5) actually has multiple results.

A phase laser ranging system $D$ with a modulated ruler measures $\frac{\lambda}{2}$ the distance of length. If the distance $D$ is between $N$ times $\frac{\lambda}{2}$ and $N + 1$ times $\frac{\lambda}{2}$ the ruler measures $N$ times, and then there is an unsatisfactory part d of $\frac{\lambda}{2}$ the distance to be measured $D = \frac{N\lambda}{2} + d$. However, due to continuous laser modulation, the system can only distin-

guish the unsatisfactory part of $\frac{\lambda}{2}$, and it is difficult to distinguish the integer times of the measurement. Therefore, only when the measurement distance does not exceed the length of the measuring ruler, that is, $N = 0$, does the distance measurement result have a unique solution.

Phase laser ranging is usually suitable for the measurement of short and medium distances, and the measurement accuracy can reach the millimeter and micron levels. It is also one of the methods with the highest ranging accuracy at present.

### 3.1.2. Pulsed Laser Ranging

Pulsed ranging is a measurement method that was first used in the field of surveying and mapping by laser technology. Due to the small laser divergence angle, the short duration of the laser pulse, and the maximum instantaneous power of more than megawatts, pulsed laser ranging has good directionality and a strong anti-interference ability. In ranging with cooperative targets, extremely long-distance measurements can be carried out, and when short-range measurements are carried out, cooperative targets are not needed and the target to be measured can also be measured by receiving the light signal of diffuse reflection. Pulse laser ranging has the characteristics of a simple system with long range, and a single measurement time that is faster than continuous laser ranging, but it also has the disadvantage of low single measurement accuracy [25].

The pulsed laser emitted by the laser is reflected after hitting the target to be measured, and the reflected light is received and amplified by the photodetector of the ranging system. Taking the transmitted optical pulse signal as the timing start time and the received optical pulse signal as the timing end time and by measuring the time between the two, the time required for the laser to travel back and forth between the ranging system and the target can be obtained. The distance to the target to be measured can be converted from the general formula of distance measurement ($S = \frac{ct}{2}$) by using the TOF method.

The pulsed laser ranging system is usually composed of several modules: the control module, laser emission module, laser connection module, receiving module, time identification module, and time interval measurement module. Figure 2 is a schematic diagram of the pulsed laser ranging system.

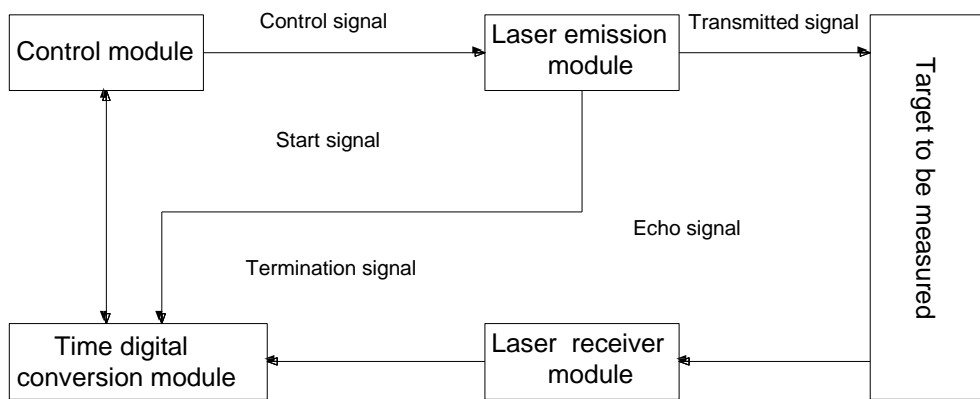

**Figure 2.** Schematic diagram of the pulsed laser ranging system.

Figure 3 is the waveform and timing diagram of the pulsed laser ranging system, where the rising edge of the signal is discriminated. The round-trip time can be detected from the number of pulses digitally measured from the recording of the time between the reference signal and the echo signal.

In the timing diagram of the pulsed laser ranging system, it can be clearly seen that the working process of the system is as follows: the pulsed laser emits laser pulses, and the laser pulses are used as emission signals. The signal reflects a small amount of energy to the receiving system as a reference signal through a half mirror and a half mirror, and most of the energy is emitted to the target through the half mirrors. The reflected light signal is received by the receiving system as an echo signal, and both the reference signal and the

echo signal are received by the photodetector and converted into electrical signals. The obtained electrical signal is shaped and amplified, the reference signal is shaped as a start signal, and the echo signal is shaped as a stop signal. The two signals are discriminated to obtain the start and end time information, and this is the time when the laser is in the ranging system. The flight time between the targets is then monitored by the time measurement method to obtain the required data.

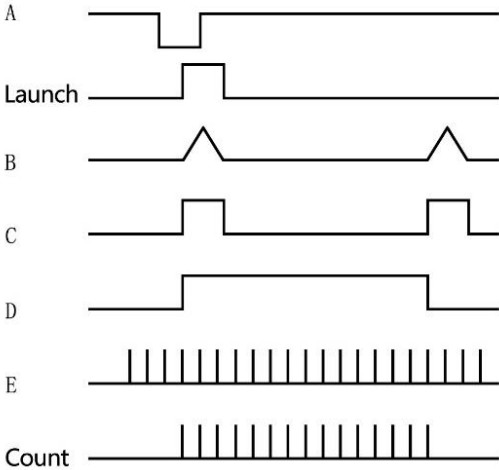

**Figure 3.** Waveform and timing diagram of the pulsed laser ranging system.

### 3.2. Measurement Principle of Wireless Laser Displacement Monitoring Equipment

Based on wireless transmission architecture, the principles of laser pulse ranging and phase ranging were combined for different application situations. Through the improvement of the algorithm principle, monitoring equipment that meets the needs of the deflection monitoring of short and medium-span bridges was developed.

A schematic diagram of the laser displacement measurement is shown in Figure 4. The laser displacement monitoring equipment is installed at the position of the bridge pier or abutment, and the reflector with enhanced radiation is installed at the measurement point. The displacement of the beam at the measuring point is calculated through the conversion of laser ranging, sensor installation angle and reflector installation angle.

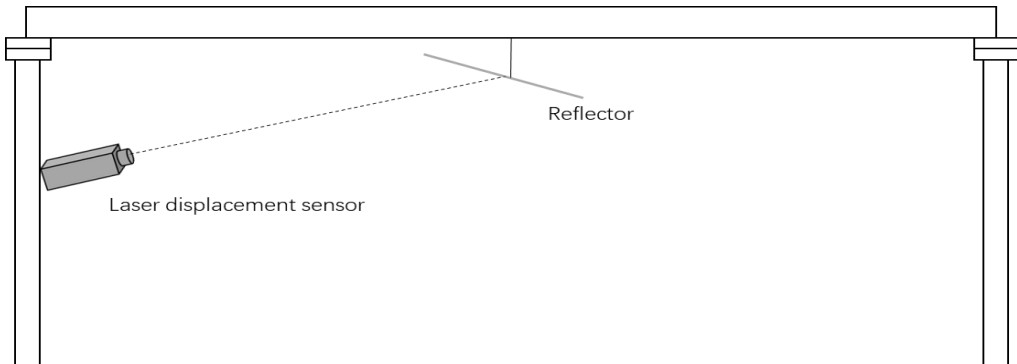

**Figure 4.** Schematic diagram of displacement measurement with laser displacement sensor.

The model of this algorithm is shown in Figure 5. The reflective device is arranged at the position where the deflection of the measured bridge is large, and the three-dimensional coordinates $(x_0, y_0, z_0)$ of the laser emission point O are read by the total station. The three-dimensional coordinates of point A (the laser reflection) are $(a_0, b_0, c_0)$, and the other three noncollinear points $A_0$ $(x_1, y_1, z_1)$, $B_0$ $(x_2, y_2, z_2)$, and $C_0$ $(x_3, y_3, z_3)$ (on the reflective device) are read as well. The laser monitoring device reads the initial distance $l_0$ from the laser monitoring device to the reflective device, and after the reflective device settles $d$, the device reads the distance between the laser monitoring device and the reflective

device $l_1$. In the model used, $\alpha$ is the initial inclination angle between the laser monitoring equipment and the horizontal plane, $\beta$ is the initial inclination angle between the reflective device and the horizontal plane, point A is the initial reflection point position of the laser monitoring equipment on the reflective device, A' is the corresponding position of the reflection point after the reflective device moved, and B is the reflection point position of the reflective device.

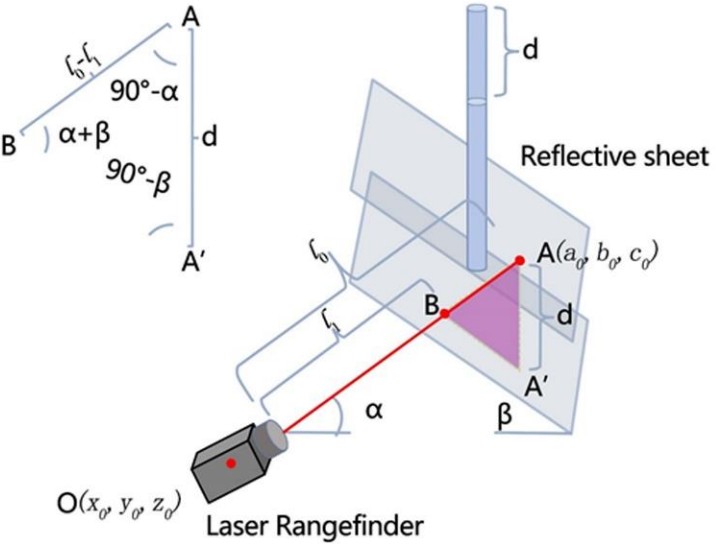

**Figure 5.** Deflection measurement model of laser monitoring equipment.

By the law of sine, we can obtain Formulas (6) and (7):

$$\frac{d}{\sin(\alpha + \beta)} = \frac{l_0 - l}{\cos \beta} \tag{6}$$

$$d = \frac{(l_0 - l)\sin(\alpha + \beta)}{\cos \beta} \tag{7}$$

The inclination angle ($\alpha$) (Formula (8)) between the laser monitoring equipment and the horizontal plane is:

$$\alpha = \arctan\left(\frac{|z_0 - c_0|}{\sqrt{(x_0 - a_0)^2 + (y_0 - b_0)^2}}\right) \tag{8}$$

According to the coordinates of the three noncollinear points ($A_0(x_1, y_1, z_1)$, $B_0(x_2, y_2, z_2)$, $C_0(x_3, y_3, z_3)$) initially read by the laser monitoring device (see Figure 6), the normal vector of the plane can be expressed as Formulas (9) and (10):

$$n = A_0B_0 \times A_0C_0 = (a, b, c) \tag{9}$$

$$
\begin{aligned}
a &= (y_2 - y_1)(z_3 - z_1) - (y_3 - y_1)(z_2 - z_1) \\
b &= (z_2 - z_1)(x_3 - x_1) - (z_3 - z_1)(x_2 - x_1) \\
c &= (x_2 - x_1)(y_3 - y_1) - (x_3 - x_1)(y_2 - y_1)
\end{aligned} \tag{10}
$$

The angle ($\beta$) between the reflective device and the horizontal plane is Formula (11):

$$\beta = \arctan\left(\frac{\sqrt{a^2 + b^2}}{|c|}\right) \tag{11}$$

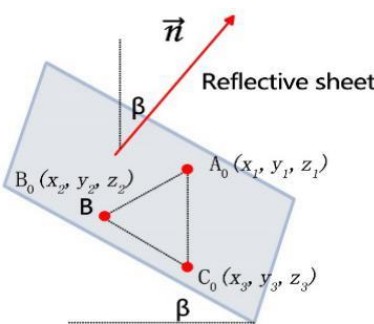

**Figure 6.** Schematic diagram of the deflection measurement of the laser monitoring equipment.

Let the error coefficient $k$ be Formula (12):

$$k = \frac{\sin(\alpha + \beta)}{\cos \beta} \tag{12}$$

Assuming that the accuracy of laser ranging is $\Delta l$ and the accuracy of displacement measurement is $\Delta d$, the displacement conversion accuracy is expressed in Formula (13):

$$\Delta d = k\Delta l \tag{13}$$

It can be seen from the above formula that the smaller the value of $k$ is, the higher the displacement accuracy of the conversion. Therefore, the smaller $\alpha$ and $\beta$ are, the smaller k and $\Delta d$ are. When $\beta$ approaches $0°$, $\cos \beta$ approaches 1. At this time when $\Delta l$ is certain, $\Delta d$ is only related to $\alpha$. The smaller $\alpha$ it is, the smaller $\Delta d$ it is, and the correlation relationship is shown in Table 7. When $\sin \alpha = 0.1$ if the distance change measured by the laser was 1 mm, it could be converted into a displacement change of 0.1 mm; when $\sin \alpha = 1$ (that is, vertical incidence), if the distance measured by the laser changed by 1 mm, it could be converted into a displacement change of 1 mm. Therefore, theoretically, the measurement accuracy of bridge deflection can be improved by using a small-angle launch method.

**Table 7.** Relationship between the theoretical deflection conversion accuracy and the laser emission angle $\alpha$.

| Serial Number | Laser Ranging Accuracy $\Delta l$ (mm) | Displacement Measurement Accuracy $\Delta d$ (mm) | $\alpha$ (°C) |
|:---:|:---:|:---:|:---:|
| 1 | 1.0 | 0.10 | 5.7 |
| 2 | 1.0 | 0.16 | 9 |
| 3 | 1.0 | 0.20 | 11.5 |
| 4 | 1.0 | 0.21 | 12 |
| 5 | 1.0 | 0.26 | 15 |
| 6 | 1.0 | 0.30 | 17.5 |
| 7 | 1.0 | 0.40 | 23.6 |
| 8 | 1.0 | 0.50 | 30 |
| 9 | 1.0 | 0.71 | 45 |
| 10 | 1.0 | 0.87 | 60 |
| 11 | 1.0 | 1.0 | 90 |

*3.3. Indoor Static Loading Test for Verification*

3.3.1. Experimental Setup

To test the feasibility and accuracy of the above algorithm model, experiments were carried out in the laboratory for verification. Since the laser monitoring equipment is only a simple data acquisition, to realize the real-time upload of data, a networking equipment—gateway was needed, so the experimental equipment included: laser moni-

toring equipment, total station, gateway, loading equipment, microprism reflective film, etc. Among them, the basic performance parameters of the laser monitoring equipment wee the initially determined requirements (shown in Tables 4–6), which meet the proposed displacement monitoring requirements for short and medium-span bridges. The incident angle range of the microprism reflective sheet was 5°~40°.

　　The indoor verification process adopted static loading. The loading test beam was a hollow plate beam with a length of 20 m and 2 tons of concentrated load on the mid-span. The measurement and comparison equipment used was a dial indicator with an accuracy of 0.001 mm. The installation of the test equipment is shown in Figure 7.

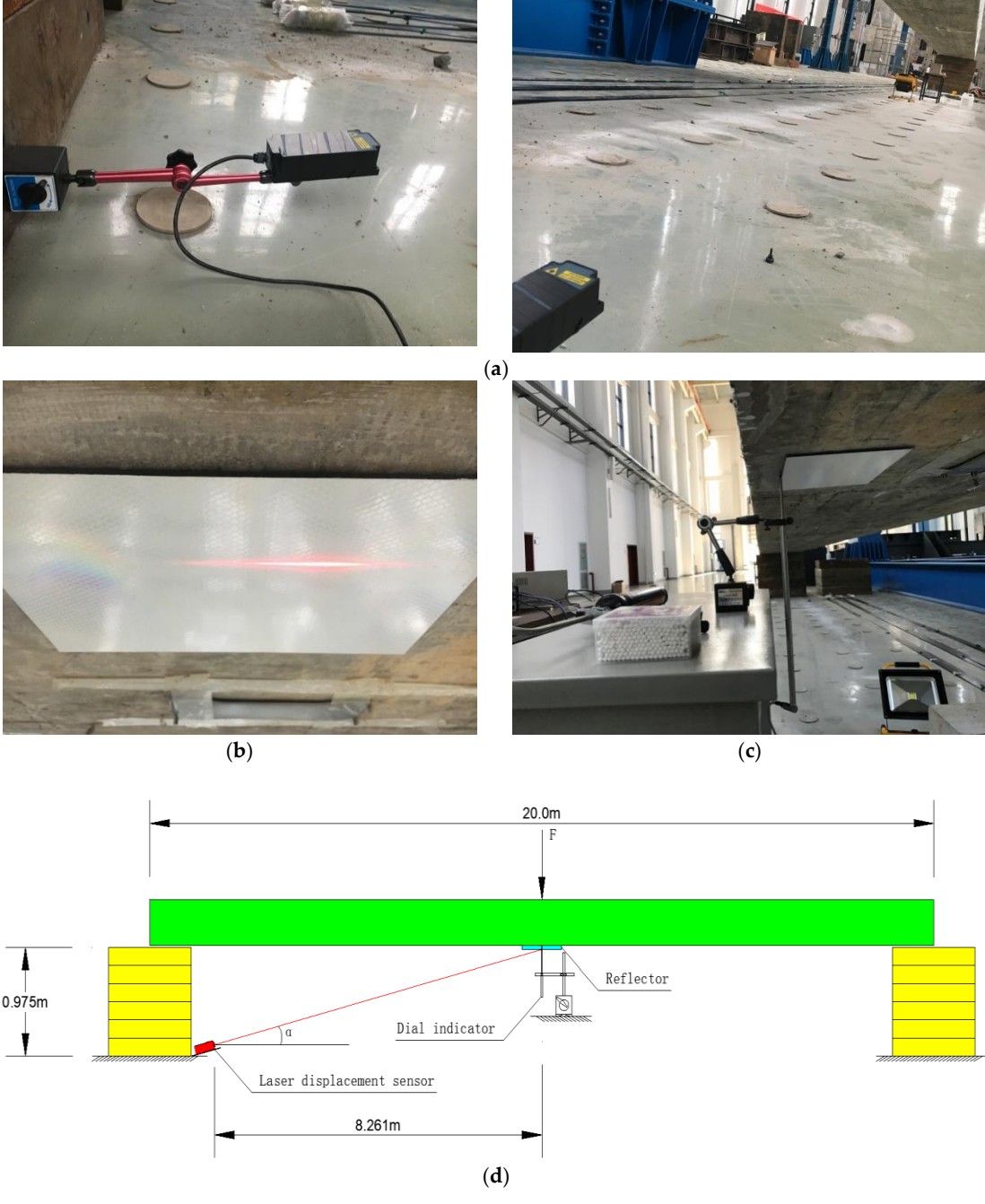

**Figure 7.** Test scene. (**a**) Installation of laser displacement monitoring equipment. (**b**) Microprism reflective sheet. (**c**) Dial indicator. (**d**) Schematic diagram of the experiment.

### 3.3.2. Analysis of Experimental Data

Before the test started, the distance measurement error of the laser displacement monitoring equipment was measured. The error calculation can be obtained through the static distance measurement test. When no load is applied, the distance measurement data can be collected for a period of time. Figure 8 shows the time–history curve of the measurement distance. The range is between 8.2946 m and 8.2966 m, and the fluctuation range is approximately 2.0 mm. Generally, the random error of measurement conforms to a Gaussian distribution. The greater the number of measurements, the closer the average value of the measurement is to the true value. The average distance value in the test period is subtracted from the measured distance value, and the probability distribution of the distance measurement error can be further obtained, as shown in Figure 9. It can be seen from the figure that the error distribution is in a certain interval near 0 ($-1.2$ mm~0.8 mm), $\Delta l_{max} \approx 2.0$ mm, and the long-term ranging error probability of the laser displacement monitoring equipment conforms to the Gaussian distribution, which can be obtained by fitting ($\mu = 0.0592$, $\sigma = 0.6052$).

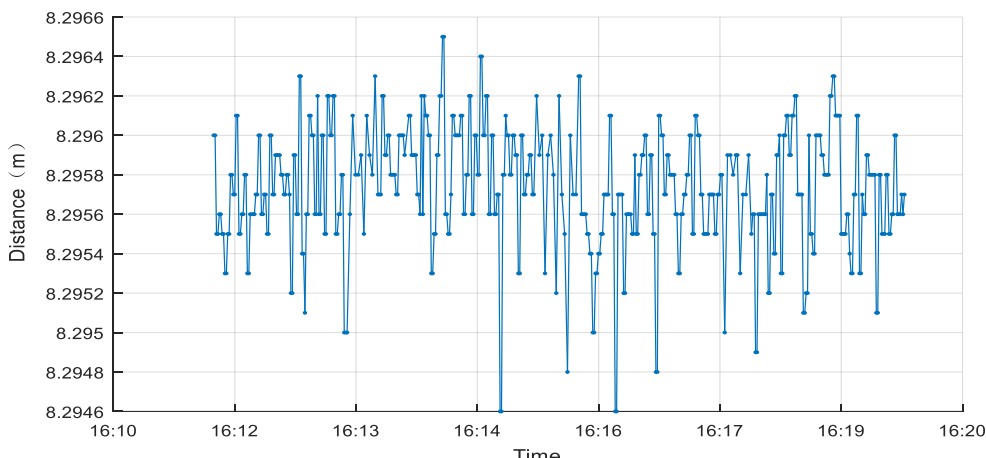

**Figure 8.** Static ranging test of laser displacement monitoring equipment.

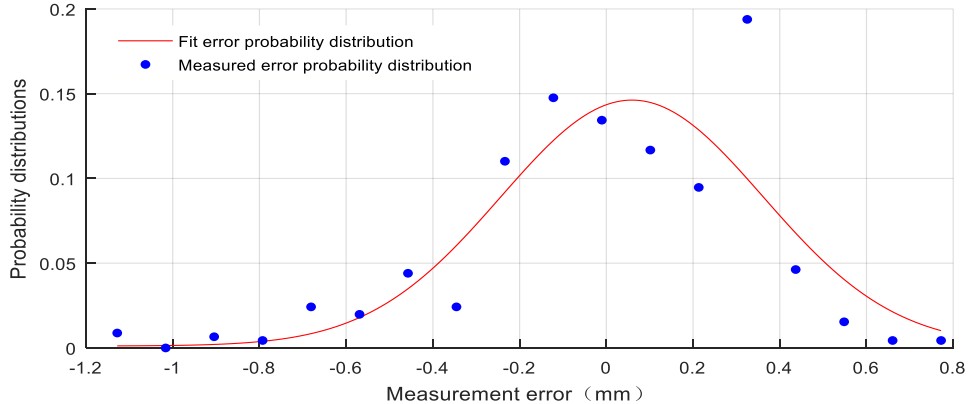

**Figure 9.** Fitting of the measurement error distribution ($a = 0.11$, $\mu = 0.0592$, $\sigma = 0.6052$).

After the static load test equipment was installed, the installation angle of the measurement sensor was approximately 5.2°, and the static average measurement distance was 8.2957 m. The deformation time–history curve of the test equipment (laser displacement monitoring equipment, red curve) and the comparison equipment (dial indicator, blue curve) taken during the test are shown in Figure 10. It can be seen from the figure that, compared with the basically flat curve of the dial indicator, the fluctuation range of the deformation measurement of the laser displacement monitoring equipment was within 0.2 mm.

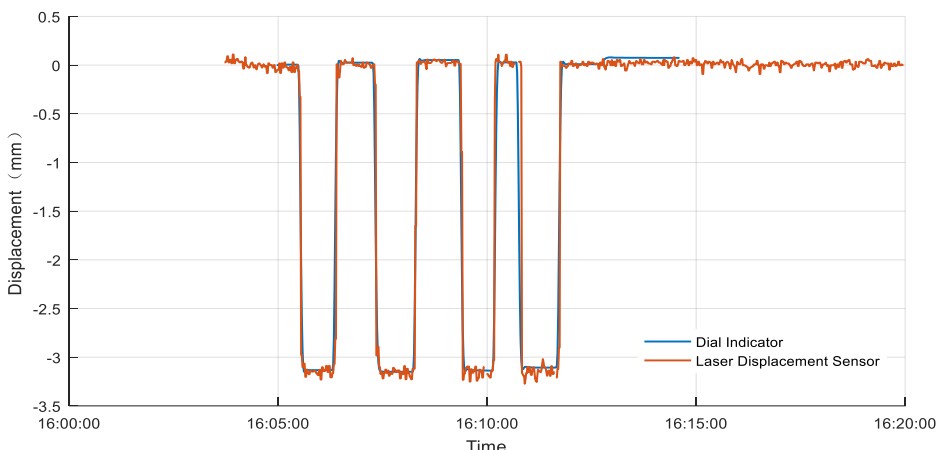

**Figure 10.** Displacement curve of the whole process from loading to unloading.

Theoretical error $\Delta d_{max}$ is calculated by the following formula:

$$\Delta d_{max} = \Delta l_{max} \cdot \sin \alpha \tag{14}$$

The maximum theoretical error obtained by calculation is 0.18 mm, and the maximum error of the measured value is relatively close to the maximum error of the theoretical value. Although this method was validated in the laboratory, on site, it may be limited by light or installation conditions and the relevant data may be uncertain, which means it requires further validation in the future.

In view of the stability of laser intensity and laser spot, the current test results show that the stable measurement results can be acquired in 25 m, and the corresponding bridge span is about 50 m.

## 4. Economic and Application Discussion

### 4.1. Economic Analysis

Research on health monitoring sensors for short and medium-span bridges and the installation of a wireless sensor network can effectively reduce the hardware cost of the health monitoring system for these types of bridges in the road network and promote the application of the health monitoring system for these bridges. Based on the bridge monitoring data center, the monitoring data can be integrated on a platform for unified analysis and processing. It can provide better data support for transportation industry authorities, bridge owners, traffic management departments, the public and bridge design researchers.

The authors have studied the displacement monitoring scheme of short and medium-span bridges and compared it to the common schemes on the market, such as the use of connecting pipes, inclinometers, satellite positioning systems, laser image methods, etc., to measure bridge displacement. Through the analysis of the hardware cost, construction cost, later maintenance cost, the reliability and performance of monitoring results, and market research a comparison of monitoring schemes was carried out.

Comparative analysis of the hardware cost showed that the displacement monitoring scheme of the laser displacement monitoring equipment developed in this paper has a lower comprehensive cost than other common schemes on the market. Because of the intelligent data acquisition method with variable frequency, the monitoring results of the laser ranging scheme studied in this paper also have strong applicability and are suitable for large-scale promotion in the displacement monitoring of short and medium-span bridges.

### 4.2. Typical Case Analysis

A three-span hollow slab girder bridge was selected. The bridge consists of eight hollow slab girders. Two of the spans are planned to be monitored. Two sets of health

monitoring programs were compared by selecting different monitoring items for their costs. Option 1 adopts only the displacement response to establish a plan for bridge health monitoring; option 2 adopts the common plan for the health monitoring of large-span bridges.

Option 1: Referring to the arrangement of deformation sensors of conventional hollow slab girder bridges, the bridge needs to be arranged with eight displacement monitors.

Option 2: The bridge needs to be equipped with eight displacement gauges and eight strain gauges, and according to the current state of the bridge, eight crack gauges should be placed, one thermometer in the middle of each lane of the two spans with one acceleration sensor and a weighing system on one side of the bridge are required, and for a total of eight displacement gauges, eight strain gauges, eight crack gauges, four acceleration sensors, four thermometers and a weighing system are needed.

(1)　The use of the plan developed in this paper for short and medium-span bridge health monitoring systems saves 43,500 Yuan RMB compared to the plan configured by commercial products, which has certain economic advantages.

(2)　Commercial sensor manufacturers in the current market generally only provide sensor equipment, installation and data receiving software, while the design of the bridge health monitoring system, the selection of measuring points, and the further analysis and structural evaluation of the data all need to be carried out by bridge professionals. Completed by a technical unit, only the cost of data analysis software is included in the commercialized system, and the cost incurred in actual operation may not be limited to this. The whole system independently developed by the research group has fully been integrated into the health monitoring system from the initial system design to the final structural state analysis with other methods. The bridge management unit no longer needs the participation of a third party, which can save communication costs and labor costs.

(3)　Since the whole system adopts the method of independent research and development, if there is any problem during the operation period of the system, it can be solved by a set of system maintenance controls. Therefore, in actual operation, it is possible to avoid the complicated problems of communication and coordination between the owner, the system design and executing company, and the hardware equipment manufacturer in the current bridge health monitoring system, and it is possible to reduce three-party communication to two-party communication, which significantly improves work efficiency.

(4)　The corresponding data receiving and analysis software can be shared by multiple bridges; if the displacement monitor and data acquisition instrument in the monitoring system can be mass-produced, the cost can be further reduced.

## 5. Conclusions

(1)　Short and medium-span bridges, which account for 90% of all bridges, need specific equipment to promote their long-term monitoring. The specific and economical monitoring system based on mid-span displacement developed here is a feasible method to help improve the monitoring system for these bridges.

(2)　For concrete, simply supported, girder bridges whose span is within 30 m, the proportion of vehicle live load in all loads decreases as the span increases, the vehicle passing time is approximately 1 s, and the mid-span deflection is approximately 15 mm.

(3)　For a span within 30 m, cheap sensors including the displacement is ± 30 mm, the accuracy is ± 0.3 mm, the resolution is 0.1 mm, and the sampling frequency is 10 Hz, with specific specifications to monitor mid-span, which can enrich the monitoring system data and improve the evaluation efficiency.

(4)　For the laser displacement sensor that integrates the principles of laser pulse ranging and phase ranging, the maximum theoretical error is calculated to be 0.18 mm, and the maximum error of the measured value is relatively close to the maximum error of the theoretical value.

(5) Because of the intelligent data acquisition method with variable frequency, these monitoring results have strong applicability and are suitable for promotion in the displacement monitoring of short and medium-span bridges as one factor with other methods for health monitoring.

(6) The self-developed data receiving and analysis software can be shared by multiple bridges; if the displacement monitor and data acquisition instrument in the monitoring system can be mass-produced, the cost can be further reduced.

**Author Contributions:** Data curation, X.L.; Formal analysis, X.L.; Funding acquisition, X.L.; Methodology, F.L.; Project administration, F.L.; Resources, X.L.; Supervision, F.L.; Validation, X.L.; Visualization, X.L.; Writing—original draft, X.L.; Writing—review & editing, F.L. All authors have read and agreed to the published version of the manuscript.

**Funding:** The Key Research and Development Program of Zhejiang Province (2021C01106); the Project of Science and Technology Program of Department of Transport, Zhejiang Province (2019050, 2020048).

**Institutional Review Board Statement:** Not applicable.

**Informed Consent Statement:** Not applicable.

**Data Availability Statement:** The data used to support the findings of this study are included within the article.

**Conflicts of Interest:** The authors declare no conflict of interest.

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
