# Peer review of "Displacement Monitoring Requirements and Laser Displacement Monitoring Technology of Bridges with Short and Medium Spans"

_applsci, doi:10.3390/app12199663_

Round 1
Reviewer 1 Report
This paper addresses the use of laser displacement monitoring systems for short and medium span concrete bridges. Specific comments that must be addressed are below.
1) A very significant overall comment is that the authors never address the fact that relying only on displacement data to assess the condition and structural capacity of a bridge – especially short and medium-span concrete bridges such as those considered here – is very limiting. Condition degradation of these bridges is often very localized, consisting of concrete cracking and rebar corrosion. This localized damage will often not result in a significant increase in displacement of these structures, which are generally quite stiff. This issue must be addressed by the authors.
2) Related to (1), the authors compare their system with a conventional system that contains strain gauges and crack gauges, and requires structural analysis, evaluation and interpretation of results by bridge engineers. It is not fair to compare a displacement-only monitoring system proposed here with a much more comprehensive system that gives significantly more comprehensive feedback on load distribution through strain gauging and damage progression through crack monitoring. It is also important to note that the assessment of data and response by a bridge engineer is a critical component to any monitoring system, and should not be bypassed regardless of monitoring system used. The authors must revise this comparison and fully acknowledge its shortcomings.
3) Related to comment (2), the statement on page 14 “… the whole system independently developed by the research group has fully covered the health monitoring system from the initial system design to the final structural state analysis” is not supported by the work presented in this paper. The authors are proposing a laser displacement measurement device, and have validated its accuracy by comparing results with traditional dial gauge measurements. There has been no attempt to use the data to assess structural condition or performance.
4) The authors must provide some details of their FE model, including element type, model fidelity, method of load application, etc. These details are need to allow the reader to independently assess the model quality.
5) Building on comments 1-3, conclusion (1) is not justified based on what is presented in the paper. It must be modified or removed following revisions per the comments above.
6) Similarly, conclusion (3) claims that the laser monitoring system “can meet the monitoring needs”, and that is not justified by the work presented here.
7) Conclusion (5) is not justified based on the unfair comparison noted above, and the lack of information on bridge response that is provided only by monitoring displacement.
Reviewer 2 Report
Following minor typographical errors be corrected:
1. Table 1 should use symbols and abbreviations. The contents should be briefly written using abbreviations.
2. From Equations1 and 2, the symbol of % is to be removed.
3. The load impact system, denoted by symbol of mu, should be defined more elaborately for the readers to grasp easily.
4. On page 5 , line 2, please add ''It can be observed'' before from.
5. In Table 6, protection class IP67 may be explained in greater detail.
6. In Fig. 3, the spelling of module is wrong. Please correct it.
7. The findings of the results should be compared and critically evaluated in the body of the paper.
Round 2
Reviewer 1 Report
The paper has been sufficiently revised per this reviewer's comments.
Author Response
We revised the manuscript again.
Additional revisions were made as suggested by the academic editor.
Many thanks for reviewers and editor.